# VIDEOGUARD: PROTECTING VIDEO CONTENT FROM UNAUTHORIZED EDITING

## ABSTRACT

With the rapid development of generative technology, current generative models can generate high-fidelity digital content and edit it in a controlled manner. However, there is a risk that malicious individuals might misuse these capabilities for misleading or unlawful activities. Although existing research has attempted to shield photographic images from being manipulated by generative models, there remains a significant disparity in the protection offered to video content editing. To bridge the gap, we propose a protection method named VideoGuard, which can effectively protect videos from unauthorized malicious editing. This protection is achieved through the subtle introduction of nearly unnoticeable perturbation that interferes with the functioning of the intended generative diffusion models. Different from images, videos consist of sequential frames, containing not only visual content but also motion dynamics. Due to the redundancy between video frames, and inter-frame attention mechanism in video diffusion models, simply applying image-based protection methods separately to every video frame can not shield video from unauthorized editing. To tackle the above challenge, rather than optimize perturbation in a frame-wise manner like image-based methods, we adopt joint frame optimization, treating all the video frames as an optimization entity. Furthermore, we extract video motion information and fuse it into optimization objectives. Thereby, these alterations can effectively compel the models to produce outputs that are implausible and inconsistent. We provide a pipeline to optimize such a perturbation. Finally, we use both objective metrics and subjective metrics to demonstrate the efficacy of our method, and the results show that the protection performance of VideoGuard is superior to all the baseline methods.

## 1 INTRODUCTION

Recently, there has been great progress on generative models(Ho et al. (2020); Croitoru et al. (2023); Song et al. (2020a); Song et al. (2020b)), and the quality of the content they create continues to improve. Current technology can generate very realistic images(Nichol et al. (2021); Ramesh et al. (2022); Rombach et al. (2022); Saharia et al. (2022)) and can be edited in a controlled manner(Zhang et al. (2023); Kim et al. (2022); Zhang et al. (2023)). With the rapid development of image generation technology, video tasks have also received more and more attention(Khachatryan et al. (2023); Wu et al. (2023); Singer et al. (2022); Brooks et al. (2024)). Nowadays, readily available open-source models, especially diffusion-based models have simplified the process of altering and modifying visual media such as photos and videos. Such technology has brought great convenience to film, television, entertainment, and other industries. Nevertheless, the easy use of these models has raised concerns about their potential abuse(Shen et al. (2024); Gu (2024)). For example, someone posts their photos or videos online, and an adversary can maliciously modify the video content to slander or create fake news(Yu et al. (2024b); He et al. (2024); Salman et al. (2023)). Such abuse poses a significant security risk to individuals and underscores the critical importance of studying protection algorithms. Several prior studies have suggested methods to protect images from unauthorized or inappropriate use by preemptively embedding adversarial perturbation(Liu et al. (2023); Salman et al. (2023)). This perturbation is carefully designed based on the diffusion models. When edit models are employed by people with bad intentions, clean images can be easily and maliciously modified, while images with protective perturbation can mislead the diffusion model and result in distorted edit content. A standout protection method is Photoguard(Salman et al. (2023)), which

Figure 1: Left: The overview of video editing protection. Right: The first row represents the DDIM inversion latent obtained from the original video; the second line represents DDIM inversion latent with the same motion prompt, we can see the motion consistent with the original video; the third row represents random latent with the same motion prompt, which results in random motion dynamic; the last row represents DDIM inversion latent with another motion prompt, and the resulting video shows the motion is consistent to the original video

efficiently blocks the functionality of latent diffusion models (LDMs), forcing them to produce unsatisfactory edits for a specific image. Furthermore, there have been parallel research efforts aimed at safeguarding certain artistic styles or elements from being exploited in the training of generative models(Zhai et al. (2023); Peng et al. (2023)). However, protecting videos from editing has been less explored in our community. Similar to the image-based edit protection setting, in the video-based edit protection task, clean videos can be easily and maliciously modified, while videos with protective perturbation can mislead the diffusion model and result in distorted edit content. Figure 1 (left) presents the video edit protection overview. One intuitive approach to video protection is to apply the previous image-based method(Salman et al. (2023); Li et al. (2024)) to each frame. However, video is a signal containing motion information. There exists high content redundancy between frames. Moreover, most video edit models employ a 3D attention mechanism(Wu et al. (2023); Qi et al. (2023); Liu et al. (2024b)) for consistency preservation. Directly applying an image-based method neglects the similarity of frames, thus adversaries can still make successful edits. Furthermore, in an image-based protection task, the main optimization objective is to search for a nearly unnoticeable perturbation that leads to distorted image content when edited by adversaries, while in a video-based protection task, video consistency is also a very important visual characteristic, and thus frame consistency should also be taken into consideration during perturbation optimization.

To raise the cost of unauthorized video editing, we propose a two-stage motion-based perturbation method. Our method extracts not only content information but also motion information and hides them from the edit model. When the perturbation is added to a video and the perturbed video is fed into edit models, the frame consistency will be disrupted, thereby leading to distorted content. Specifically, given a video $\mathcal{V}$ and its inversion latent $\mathcal{Z}_0$, in stage 1, we formulate an optimization problem of minimizing motion loss and content loss to seek an inversion latent $Z_{latent}$ in $\epsilon$-spherical neighborhoods of $\mathcal{Z}_0$, and then a projection gradient descent method(Madry (2017)) is provided to solve the problem. In stage 2, we regard this inversion latent as a pseudo label and formulate another optimization problem of minimizing the loss between the current inversion latent and the target inversion latent(pseudo label), and then the Particle Swarm Optimization method(Kennedy & Eberhart (1995)) is adopted to search for the best video perturbation. Fused with this protective video perturbation, we can obtain an immunized video. Consequently, when doing an editing task with the immunized video, the adversary will obtain a manipulated inversion latent from this perturbed video, and this inversion latent will be fed into the denoise process, resulting in a distorted video that can be easily perceived as fake. During the inversion latent optimization process, we treat all frames' inversion latent as a whole entity and optimize jointly with the projection gradient descent method. Through this approach, we can take inter-frame consistency into consideration, which can effectively destroy the original video's motion information. Furthermore, we use the perturbation vector searching method to optimize video perturbation in the video pixel space, which requires

fewer computation resources. As such, videos with our protective perturbation can hinder the efforts of video diffusion models and prevent malicious modification.

We choose pioneer video editing works including Tune-A-Video(Wu et al. (2023)), Fate-Zero(Qi et al. (2023)), and Video-P2P(Liu et al. (2024b)), and conduct extensive experiments on subsets of the DAVIS dataset(Wang et al. (2019)) and some real-world videos. We choose Random Noise Perturbation and Image-based Perturbation as baseline methods to demonstrate our method's efficacy and superiority. Compared to these baseline methods, our video protection method performs better, with the frame consistency metrics dropping from 90.91 to 81.55 and text-frame alignment dropping from 18.50 to 8.46 on average. In the VBench(Huang et al. (2024)) evaluation, our proposed method demonstrates superior performance compared to the baselines across five video quality-related metrics. Notably, Subject Consistency exhibited a reduction from 89.45 to 79.08, while Motion Smoothness decreased from 89.82 to 80.73, underscoring the efficacy of our approach. Moreover, VideoGuard has a more significant visual effect, which means the edit result of immunized video is severely distorted and can be easily perceived as fake, thus achieving the goal of protecting video from unauthorized editing. Contributions of our work can be summarized as follows:

- We analyze the motion characteristic in video data and introduce a motion-based video editing protection method. To our knowledge, this is the first editing protection method tailored for video editing tasks rather than directly applying image-based methods in a frame-wise manner.

- We propose a novel two-stage protection pipeline for diffusion-based video edit protection. We formulate optimization problems for latent perturbation and video perturbation respectively, and we also provide an effective gradient-based algorithm at stage 1 and an efficient gradient-free algorithm at stage 2 to solve the optimization problems. Added with nearly imperceptible video perturbation, immunized videos can mislead the diffusion models and lead to distorted edit results.

- We conduct a lot of experiments to evaluate VideoGuard. The results show that our method can effectively protect video from unauthorized editing. Compared to baseline methods, VideoGuard is superior in both qualitative and quantitative evaluations.

## 2 RELATED WORK

### 2.1 LDMs BASED VIDEO EDITING

Generative models have demonstrated impressive performance in image editing, with approaches ranging from GANs(Gal et al. (2022); Park et al. (2019); Wang et al. (2018)) to diffusion models(Avrahami et al. (2022); Kawar et al. (2023)). Inspired by image editing techniques, an increasing amount of research is dedicated to transforming latent diffusion models (LDMs) into zero-shot image editors(Liew et al. (2023); Wu et al. (2023); Qi et al. (2023)), achieving significant advancements. These strides have sparked interest in the realm of video editing, which is a specialized task within the broader field of video generation. Unlike the standard video generation process that relies solely on conditional prompts to direct the creation, video editing necessitates both a source video and a guiding prompt(Geyer et al. (2023); Wang et al. (2023)). Given an edit prompt, the attributes of the reference video can be manipulated including shape, style, and scene(Qi et al. (2023)). In addition to appearances, videos are also characterized by the motion dynamics of subjects and camera movements across frames. Recently, the idea of customizing the motion with given reference videos has also been emerging and evolving rapidly(Zhao et al. (2023); Jeong et al. (2024)). They extract the motion pattern from the original video and transfer it to a new video generation process, thus they can obtain a new video that possesses the same motion pattern.

Editing a video essentially involves manipulating a sequence of images arranged sequentially in time. Yet, ensuring consistency across the frames of the edited video remains a challenging issue. Directly altering each frame of the source video might result in inconsistencies such as varying backgrounds or altered positions of the foreground objects(Shin et al. (2024); Ceylan et al. (2023)). The latest video editing frameworks leverage various pre-trained LDMs, to maintain frame-to-frame consistency, these frameworks commonly incorporate cross-frame global attention mechanisms(Wu et al. (2023); Yang et al. (2023)). Additionally, some systems incorporate supplementary conditions like depth, pose, and edge data to further enhance consistency(Khachatryan et al. (2023)). Given the

proliferation of sophisticated techniques for producing high-fidelity videos, there is a rising concern that these editing tools could be exploited for nefarious purposes, potentially leading to the creation of videos that are unlawful, deceptive, or damaging. In light of this, we are pioneering research in this domain.

## 2.2 Protection Against LDMs-based Digital Content Editing

Latent diffusion models (LDMs) can edit images and videos based on conditional prompts, which can potentially be exploited to generate malicious content(Chen et al. (2023); Liu et al. (2024a)). To counter this threat, Photoguard(Salman et al. (2023)) has been introduced as a protective measure for images, aiming to hinder the efforts of LDMs. This method incorporates adversarial perturbations into images, effectively perplexing LDMs and preventing unauthorized editing. Furthermore, LDMs can quickly learn specific objects or artistic styles through personalized techniques like DreamBooth(Ruiz et al. (2023)). To protect intellectual property or portrait rights, some works add perturbation to images before releasing them on the Internet(Peng et al. (2023)). With such perturbed images, the fine-tuned LDMs are only capable of producing low-quality results. Recently, there has been great advancement in video generation and video editing techniques, though brought convenience, the threat of malicious video editing is being taken into consideration. However, video edit protection has been less explored in the community. One intuitive protection method is to transfer the image-based method to a video protection task(Li et al. (2024)), applying the image-based method to every single frame. Nevertheless, this method ignores the relation between video frames. Frames' content redundancy and inter-frame attention mechanism in video diffusion models convey content information and motion information, which indicates that frame-wise protection manner will not have good protection performance.

## 3 Method

### 3.1 Threat Model

Firstly, we will clarify the threat model from the attacker's and protector's perspective.

**Attacker's Capability and Goal.** Individuals with malevolent motives have the capability to effortlessly procure a pre-trained video editing model and make alterations to videos of targeted individuals. As a result of the open-source characteristics of the LDM models and the ready accessibility of videos, perpetrators can falsify identities or fabricate misinformation, subsequently leading to potentially detrimental consequences.

**Protector's Capability and Goal.** Protectors are merely endowed with access to the video, thereby confining their operations solely to manipulating the visual content. Their task involves introducing imperceptible perturbations to the original video. The primary objective of the protector is to raise the cost of modifying the safeguarded video, thereby impeding video editing models from easily altering the video content.

### 3.2 Adversarial Perturbation

For a given computer vision model and an image, an adversarial example is an imperceptible perturbation of that image that manipulates the model's behavior(Goodfellow (2014)). In image classification, for example, an adversary can construct an adversarial example for a given image x that makes it classified as a specific target label $y_{target}$. This construction is achieved by minimizing the loss of a classifier $f_\theta$ for that image:

$$\delta_{adv} = \underset{\delta \in \Delta}{argmin} \mathcal{L}(f_\theta(x + \delta), y_{target}). \tag{1}$$

Here, $\Delta$ is a set of perturbations that are small enough that they are imperceptible-a common choice is to constrain the adversarial example to be close to the original image, i.e., $\nabla = \{\delta : ||\delta||_p \leq \epsilon\}$. The canonical approach to constructing an adversarial example is to solve the optimization problem (1) via projected gradient descent(Madry (2017)).

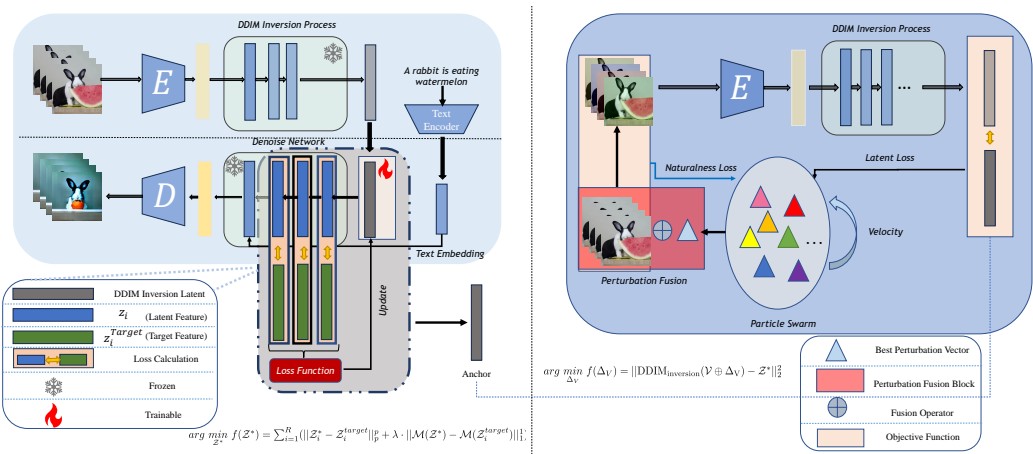

Figure 2: VideoGuard Pipeline. The left shows the diagram of optimization stage 1, we manipulate the denoise process and obtain latent perturbation. The right shows the search process of stage 2. Using latent perturbation obtained in stage 1, we can use the Particle Swarm Optimization method to seek a video perturbation.

### 3.3 MOTIVATION AND OVERVIEW

**Motivation.** Motivated by Jeong et al. (2024), we conducted some experiments on the inversion latent. As shown in Figure 1 (right), once we get the inversion latent from a video with a skiing motion pattern, this inversion latent will be fed into the denoising process guided by the target prompt and obtain edited video with skiing motion. At the same time, when we use the same inversion latent and the 'running' prompt, we can get a video with the dog running. More precisely and importantly, the dog's motion pattern is still consistent with the original one. However, a random inversion latent generates a random motion pattern as illustrated on the fourth row in Figure 1 (right). The inversion latent contains a precise motion pattern of the original video. This observation motivates us that we may spare more attention to inversion latent in video edit protection tasks. What's more, it indicates that we should regard the video and its inversion as a whole.

**Protect Pipeline Overview.** The video editing process consists of two stages. Firstly, the initial noise latent is obtained from the original video in the first stage through the DDIM inversion process. Then, the start noise will be fed into the denoising process guided by target edit prompt $\mathcal{P}$. Based on this edit pipeline, we propose a corresponding protection pipeline. Figure 2 shows the pipeline. To be specific, firstly, we optimize an initial inversion latent as a pseudo label, which has content and motion distortion compared to the original initial inversion latent. Then we regard it as an anchor in the following video space optimization. In the second stage, we employ the gradient-free PSO algorithm(Kennedy & Eberhart (1995)) to search for a perturbation in video space that can make the DDIM inversion latent obtained from the perturbed video close to the target initial inversion latent. Thus, the perturbation in video space we found can affect the start point of the editing pipeline's denoising process, and finally disrupt the edit video generation.

### 3.4 A TWO-STAGE PROTECTION PIPELINE

Video editing is aimed at using edit prompt $\mathcal{T}$ to generate a new video $\mathcal{V}_{edit}$ derived from a given source video $\mathcal{V}$. The whole process can be formulated as $\mathcal{V}_{edit} = Edit(\mathcal{V}, \mathcal{T})$, and it can be divided into two parts: (1) the initial inversion latent is obtained through DDIM inversion process; (2) the edited video is obtained through denoise process guided by the edit prompt. Given a video $\mathcal{V} = [\mathbf{x_1}, ..., \mathbf{x_n}]$ with n frames, the start noise latent $\mathcal{Z} = [\mathbf{z_1}, ..., \mathbf{z_n}]$ can be obtained through the DDIM inversion process, which can be formulated as

$$\text{Inversion} : \mathcal{V} \rightarrow \mathcal{Z}, \quad \mathcal{Z} = \text{DDIM}_{\text{inversion}}(\mathcal{E}(\mathcal{V})). \tag{2}$$

With initial noise latent $\mathcal{Z}$, we can get the edited video $\mathcal{V}_{edit}$ through denoise process guided by edit prompt $\mathcal{T}$

$$\text{Denoise} : \mathcal{Z} \rightarrow \mathcal{V}_{\text{edit}}, \mathcal{V}_{\text{edit}} = \mathcal{D}(\text{DDIM}_{\text{sample}}(\mathcal{Z}, \mathcal{T})). \tag{3}$$

Here, $\mathcal{E}$ and $\mathcal{D}$ are the VAE encoder and decoder respectively. To raise the cost of editing a video, our goal is to find such a perturbation $\delta_{video}$ that when the perturbation is added to the source video $\mathcal{V}$, it will mislead LDMs' functionality and prevent the model editing the source video successfully. At the same time, the perturbation should be as imperceptible as possible. We annotate the perturbed video as $\mathcal{V}^*$, namely $\mathcal{V}^* = \mathcal{V} + \delta_{video}$. Mathematically, our goal can be formulated as

$$arg \max_{\delta_{video} \leq \epsilon} \text{Distance}(\text{Edit}(\mathcal{V}, \mathcal{T}), \text{Edit}(\mathcal{V} + \delta_{\text{video}}, \mathcal{T})), \quad (4)$$

where Distance can be $l_2$ norm, PSNR, etc. To tackle the optimization problem above, we propose a two-stage method to search for a video perturbation. Specifically, we decompose the whole optimization objective into two parts. Firstly, we use the gradient descent method to optimize a DDIM inversion latent as an anchor; then we use the PSO algorithm to find the final video perturbation.

**Stage 1: Optimize DDIM Inversion Latent.** In stage 1, we optimize an inversion latent that can result in distorted video content after the denoise process. Let's consider the denoise process Denoise : $\mathcal{V}_{\text{edit}} = \mathcal{D}(\text{DDIM}_{\text{sample}}(\mathcal{Z}, \mathcal{T}))$. In this stage, our target is to find an optimized latent $\mathcal{Z}^*$ that can generate video $\mathcal{V}^*_{edit}$ with distorted content through a denoise process guided by the same prompt. Our problem can be formulated as follows

$$arg \max_{\delta_{latent} \leq \epsilon} \text{Distance}(\mathcal{D}(\text{DDIM}_{\text{sample}}(\mathcal{Z} + \delta_{\text{latent}}, \mathcal{T})), \mathcal{D}(\text{DDIM}_{\text{sample}}(\mathcal{Z}, \mathcal{T}))). \quad (5)$$

Briefly speaking, we search for an initial latent $\mathcal{Z}^* = \mathcal{Z} + \delta_{latent}$ in the $\epsilon$-neighborhood of the original latent $\mathcal{Z}$. Problem (5) has the same formulation as (1), thus can be solved by adversarial attack techniques(Madry (2017)). Notably, there are $T$ denoise steps in the DDIM sample process, so there will be $T$ intermediate latent features. We denote the i-th step feature as $\mathcal{Z}_i$, thus $\mathcal{Z} = [\mathcal{Z}_1, ..., \mathcal{Z}_n]$. It is intangible to measure the distance between $\mathcal{V}^*_{edit}$ and $\mathcal{V}_{edit}$ because, we do not know the edit prompt and the edit model, i.e. we can not get the edited video $\mathcal{V}_{edit}$. To quantify the objective in problem (5), we follow the strategy mentioned in PRIME(Li et al. (2024)) and PhotoGuard(Salman et al. (2023)). Namely, rather than maximize the difference between the edited video derived from the original one and the protected one, we choose target latent features $\mathcal{Z}^{target}_i$ to guide the optimization direction at every step. $\mathcal{Z}^{target}_i$ can be any random latent, and we set $\mathcal{Z}^{target}_i$ zero in our experiment setting. As stated in Section 3.2, we can use PGD to solve such an optimization problem:

$$arg \min_{\delta_{latent} \leq \epsilon} \text{Distance}(\mathcal{D}(\text{DDIM}_{\text{sample}}(\mathcal{Z} + \delta_{\text{latent}}, \mathcal{T})), \mathcal{D}(\text{DDIM}_{\text{sample}}(\mathcal{Z}_{\text{target}}, \mathcal{T}))). \quad (6)$$

In previous works, the perturbation is optimized in a frame-wise manner. Nevertheless, there exists content redundancy between video frames and the 3D mechanism in video diffusion models can propagate visual features among frames to maintain consistency when doing generation. The ignorance of these characteristics in frame-wise optimization will lead to obtaining perturbation with no effect. Consequently, we propose to seek a perturbation by treating the video inversion latent as a single vector. Specifically, for the latent $\mathcal{Z}_i = [\mathbf{z}_{i,1}, ..., \mathbf{z}_{i,n}]$ at i-th denoise step, rather than optimize each $\mathbf{z}_{i,k}, k = 1, ..., n$ separately, we regard $\mathcal{Z}_i$ as a whole variable to optimize, namely optimize the whole video latent but not every separate frame image latent. Moreover, video frames can be regarded as image time series, which means we can use the first-order difference to represent the motion information of the video. As mentioned in Section 3.3, we can inject motion information during optimization. To be concrete, we annotate $\mathcal{M}(\mathcal{Z}_i) = [\mathbf{z}_{i,2} - \mathbf{z}_{i,1}, ..., \mathbf{z}_{i,n} - \mathbf{z}_{i,n-1}]$ as motion vector to represent video consistency information. Thus, our objective can be formulated as

$$arg \min_{\mathcal{Z}^*_0} f(\mathcal{Z}^*_0) = \sum_{i=1}^{T} (||\mathcal{Z}^*_i - \mathcal{Z}^{target}_i||^p_p + \lambda \cdot ||\mathcal{M}(\mathcal{Z}^*_i) - \mathcal{M}(\mathcal{Z}^{target}_i)||^1_1)$$

$$s.t. ||\mathcal{Z}^*_0 - \mathcal{Z}_0||^2_2 < \epsilon. \quad (7)$$

Note that $\mathcal{Z}^*_i$ is derived from $\mathcal{Z}^*_0$ through the denoise process, thus the objective function in (7) actually has only one optimization variable. In problem 7, we call the first term content loss and the second term motion loss. $\lambda$ is a trade-off between content loss and motion loss. We will discuss the function of $\lambda$ in the experiment section. Furthermore, the above T steps optimization process will cost unaffordable computation resources. Prompt2Prompt(Hertz et al. (2022)) and MFA(Yu et al. (2024a)) observe that the initial denoise steps are vital to the whole denoise process. To achieve joint

Figure 3: Video editing results. Real-world scenario editing protection results.

optimization, we select the first $R$ steps for loss calculation. Thus, our final optimization problem is formulated as follows:

$$arg \min_{\mathcal{Z}_0^*} f(\mathcal{Z}_0^*) = \sum_{i=1}^{R} (||\mathcal{Z}_i^* - \mathcal{Z}_i^{target}||_p^p + \lambda \cdot ||\mathcal{M}(\mathcal{Z}_0^*) - \mathcal{M}(\mathcal{Z}_i^{target})||_1^1)$$

$$s.t. \ ||\mathcal{Z}_0^* - \mathcal{Z}_0||_2^2 < \epsilon. \tag{8}$$

**Stage 2: Optimize Video Perturbation.** The second part of our proposed method is to search for a subtle perturbation in video space that can deviate its DDIM inversion latent from the source to the one we optimized in part 1. In other words, we regard the optimized latent in stage 1 as the anchor and search for video perturbation $\delta_{video}$ with the constraint $\delta_{video} \leq \epsilon$. The problem can be formulated as follows:

$$arg \min_{\delta_{video} \leq \epsilon} \mathcal{L}(\text{DDIM}_{\text{inversion}}(\mathcal{E}(\mathcal{V} + \delta_{\text{video}})), \mathcal{Z}_{\text{anchor}}). \tag{9}$$

However, it is an optimization problem in video pixel space, which means it will cost tremendous computation resources to use a gradient algorithm to find a satisfying perturbation $\delta_{video}$. To tackle this problem, we propose to use the classical optimization method PSO to find such perturbation. Rather than directly searching in the video pixel space $\mathcal{R}^{F \times C \times H \times W}$, we search for a perturbation vector $\Delta_V \in \mathcal{R}^{F \times M}$, and fuse it into the original video. We annotate it as $\mathcal{V}^* = \mathcal{V} \bigoplus \Delta_V$. We choose $\mathcal{L} = ||\text{DDIM}_{\text{inversion}}(\mathcal{V}^*) - \mathcal{Z}_0^*||_2^2$ as the objective function, so the optimization problem in this stage can be formulated as follows:

$$arg \min_{\Delta_V} f(\Delta_V) = ||\text{DDIM}_{\text{inversion}}(\mathcal{E}(\mathcal{V} \oplus \Delta_V)) - \mathcal{Z}_0^*||_2^2$$

$$s.t. \ ||\mathcal{V}^* - \mathcal{V}||_2^2 \leq \epsilon, \quad \mathcal{V}^* = \mathcal{V} \oplus \Delta_V. \tag{10}$$

Furthermore, to make our protection more realistic and imperceptible, we can constrain the naturalness when doing optimization. More details can be found in Appendix E.

## 4 EXPERIMENTS

### 4.1 IMPLEMENTATION DETAILS

Following previous works in video editing, we evaluate our method on DAVIS(Wang et al. (2019)) videos, with various text prompts on each video to obtain diverse editing results. To further demonstrate the performance of our method, we download 20 videos from the internet for real-world video editing protection evaluation. Our evaluation dataset comprises 80 text-video pairs. The spatial resolution of the videos is $512 \times 512$ pixels, and every video is composed of 8 frames. More detailed settings are listed in the Appendix A. We choose 2 methods for baseline method comparison: Photoguard(Salman et al. (2023)) applied in a frame-wise manner(namely PRIME(Li et al. (2024))), and random noise perturbation. As for video editing models, we experiment with Fatezero(Qi et al. (2023)), Tune-A-Video(Wu et al. (2023)), and Video-P2P(Liu et al. (2024b)), which are pioneer works in video editing tasks.

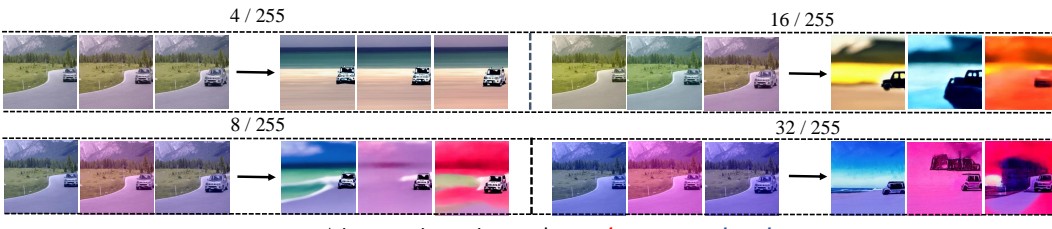

Figure 4: Different perturbation budges' impact on protection efficacy.

Table 1: Quantitative metric results. Vbench(Huang et al. (2024)) is a video evaluation benchmark, and clip image-text similarity and image-image similarity are used to represent text alignment and frame consistency.

| Method | VBench↓ | | | | | Frame-Con↓ | Text-Align↓ |
| | Aesthetic Quality | Subject Consistency | Background Consistency | Motion Smoothness | Imaging Quality | | |
| --- | --- | --- | --- | --- | --- | --- | --- |
| w/o protection | 55.93 | 89.45 | 92.33 | 89.82 | 54.53 | 90.91 | 18.50 |
| Random Noise | 56.37 | 91.40 | 93.12 | 92.53 | 52.69 | 92.33 | 15.48 |
| PhotoGuard (Salman et al., 2023) | 56.12 | 90.83 | 92.94 | 91.41 | 53.34 | 91.82 | 17.68 |
| VideoGuard(Ours) | **53.20** | **79.08** | **87.10** | **80.73** | **51.52** | **81.55** | **8.46** |

## 4.2 BASELINE COMPARISONS

**Qualitative Evaluation.** Figure 5 shows some qualitative comparison results. As shown in the figure, videos protected by Photoguard and random noise can still be edited with consistent content. Applying an image-based protection method in a frame-wise manner doesn't perform well. One of the main reasons may be that the image-based method does not consider the motion dynamics and photoguard targets for image-based diffusion models, namely, it cannot corrupt the inflation attention mechanism in video diffusion models. When the videos are protected by our method, the perturbation optimized by our method can disrupt the normal edit process and lead to content distortion. This figure also shows the editing protection effect of videos with different editing prompts. It is shown that our method is not only effective for a certain specified video-text pair but also has a good protection effect under different editing texts with a given video, which demonstrates the transferability across different prompts. Meanwhile, Figure 6 and results in Appendix F.1 present the transferability across different models.

To demonstrate real-world video editing protection efficacy, we also conducted experiments on real-world videos. Figure 3 shows the effectiveness of our method when applied in real-world scenarios.

**Quantitative Evaluation.** For quantitative evaluation, we calculate the average frame-wise clip scores for text alignment, and we adopt the clip similarity of two subsequent frames to evaluate the video's consistency. Furthermore, for a more comprehensive evaluation, we use video evaluation benchmark VBench(Huang et al. (2024)) to evaluate the edited videos at the dimensions of Aesthetic Quality, Subject Consistency, Background Consistency, Motion Smoothness, and Imaging Quality. Table 1 shows the quantitative results. As the table shows, our method has superior performance to the baseline methods, especially with Subject Consistency dropping from 89.45 to 79.08, and Motion Smoothness dropping from 89.92 to 80.73. To further demonstrate the effectiveness of our method, we use human study for auxiliary evaluation, and we can conclude from the results that our method has a more powerful protection ability. More details can be found in the Appendix D.

## 4.3 PERTURBATION BUDGET ANALYSIS

We analyze the impact of the perturbation budget on protection performance. As we can imagine, the bigger the perturbation added, the more corruption there will be. However, a bigger perturbation means the protection has a worse stealthiness. As Figure 4 shows, we can obtain a pretty well-corrupted edit result with a big perturbation budget of 32/255, while nearly no effect with a small budget of 4/255. An empirically promising budget will be between 8/255 and 16/255, with good protection functionality and a bearable degree of stealthiness.

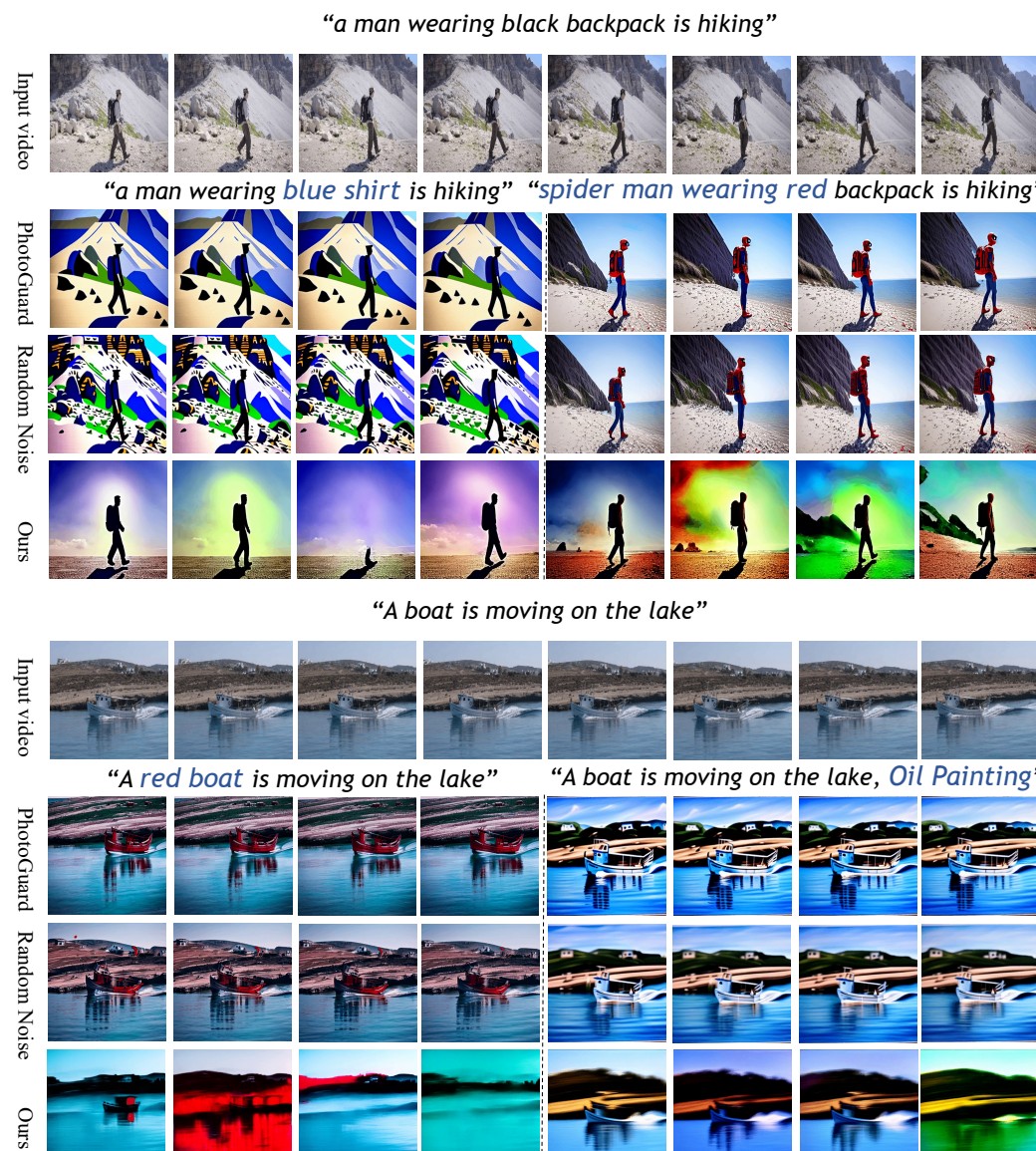

Figure 5: Video editing results. The first row represents the original video, and the following 3 rows represent Photoguard, Random Noise, and our method respectively.

We also conduct experiments to analyze the trade-off $\lambda$ between content loss and motion loss in (7) and the perturbation vector searching method used in stage 1. More details are listed in the Appendix E.

## 5 DISCUSSION

We present a novel framework for video editing protection powered by diffusion models. We study the motion dynamics of a video in the diffusion latent space and construct optimization objectives accordingly. We propose joint optimization in latent space to tackle the challenge of frame information redundancy, and we suggest applying a gradient-free search algorithm for video perturbation optimization in a small vector space rather than in the raw pixel space, which is computationally effective. Our method outperforms existing baselines and demonstrates its transferability between different models and instructions.

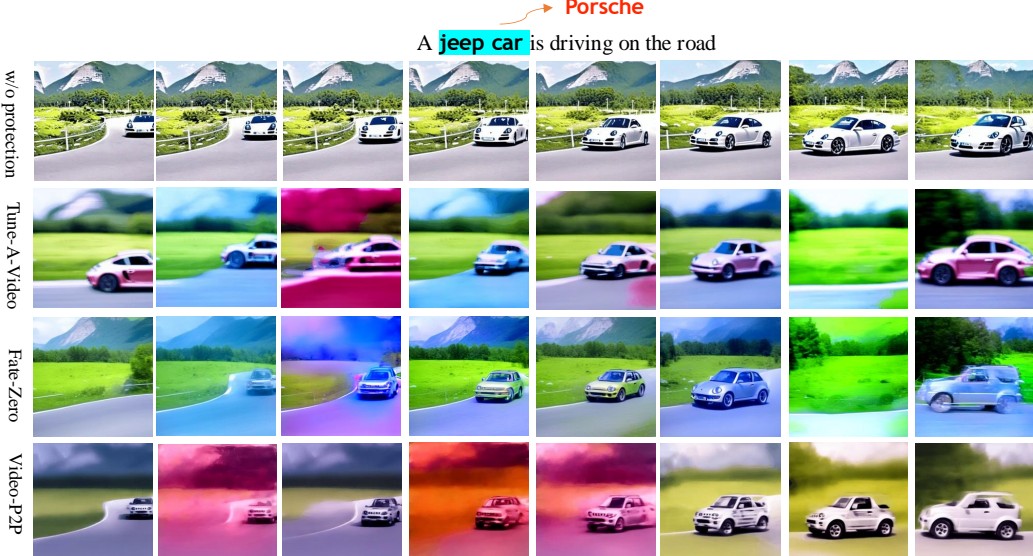

Figure 6: Transferability across different models. The first row presents video editing results that are without protection. The following 3 rows show the editing results by different editing models.

Despite the promising results, our study also has some limitations. Our method does not perform well when applied to models with additional conditional information. The main reason might be that when more and more control conditions are fused into the editing process, the importance of the source video downgrades gradually, namely the downgrade of protectors' capacity, thus downgrading the performance of the protection. A potential solution is to incorporate the conditional control module of the editing model into the perturbation optimization process, and thus the resulting perturbation can interfere with the editing model's proper reception of the control conditions. This perturbation subsequently affects the overall editing performance, thereby achieving the desired protective objective. This avenue of research is left as future work.

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

## A  IMPLEMENTATION DETAILS

**Stable Diffusion.** We use Stable Diffusion as a text-to-image model; we use stableDiffusion-v-1-5 provided via the official Huggingface web page. As for model inflation, we adopt the mechanism used in tune-a-video(Wu et al. (2023)).

**DDIM inversion.** In all of our experiments, we use DDIM deterministic sampling with 50 steps. For inverting the video, we follow Tumanyan et al. (2023) and use DDIM inversion with a classifier-free guidance scale of 7.5, and the inversion step is 50.

**Hyper-parameters.** In equation 8, we set $\lambda$ to 1, but in the experiments, the value can range from 1 to 10, which means different trade-offs are tested to get a better performance. In the first stage, we set the $\delta_{latent}$ to 10/255 while in the second stage, we set the $\delta_{video}$ to 8/255.

In the objective functions,

$$arg \min_{\Delta_V} f(\Delta_V) = ||\text{DDIM}_{\text{inversion}}(\mathcal{E}(\mathcal{V} \oplus \Delta_V)) - \mathcal{Z}_0^*||_2^2$$

$$s.t. \ ||\mathcal{V}^* - \mathcal{V}||_2^2 \leq \epsilon, \quad \mathcal{V}^* = \mathcal{V} \oplus \Delta_V, \tag{11}$$

we set parameter p to 1 when calculating motion loss, and set parameter p to 2 when calculating content loss.

**Particle Swarm Optimization.** We use PSO algorithm(Kennedy & Eberhart (1995)) to search for $\delta_{video}$. We initialize the number of the particles to 40, and the total iteration is 40.

## B    STABLE DIFFUSION.

Diffusion models have emerged recently as powerful tools for generating realistic images. These models excel especially at generating and editing images using textual prompts and currently surpass other image generative models such as GANs in terms of the quality of produced images(). Distinct from traditional diffusion models, Stable Diffusion(Rombach et al. (2022)) functions within a low-dimensional latent space, which is accessed via VAE autoencoder((Kingma (2013))) $\mathcal{E}, \mathcal{D}$. Specifically, once the latent representation $z_0$ is obtained by compressing an input image $f \in \mathbb{R}^{H \times W \times 3}$ through the encoder $\mathcal{E}$, i.e. $z_0 = \mathbf{E}(f)$, diffusion forward process gradually adds Gaussian noise to $z_0$ to obtain $z_t$ through Markov transition with the transition probability:

$$q(z_t|z_{t-1}) = \mathcal{N}(z_t; \sqrt{1 - \beta_t} z_{t-1}, \beta_t I), \quad t = 1, ..., T, \tag{12}$$

where the noise schedule $\{\beta_t\}_{t=1}^T$ is an increasing sequence of t and T is the number of diffusion timesteps. Then, the backward denoising process is given by the transition probability:

$$p_\theta(z_{t-1}|z_t) = \mathcal{N}(z_{t-1}; \mu_\theta(z_t, t), \sigma_t^2 I), \quad t = T, ..., 1. \tag{13}$$

Here, the mean $\mu_\theta(z_t, t)$ can be represented using the noise predictor $\epsilon_\theta$ which is learned by the minimization of the MSE loss with respect to $\theta : \mathbb{E}_{f,\tau,\epsilon \sim \mathcal{N}(0,I),t}||\epsilon - \epsilon_\theta(z_t, t, \tau)||_2^2$, where $\epsilon$ refers to the zero mean unit variance Gaussian noise vector, and $\tau = \Phi(\mathcal{T})$ is the embedding of a text $\mathcal{T}$. Specifically, a prevalent approach in diffusion-based image editing is to use the deterministic DDIM scheme to accelerate the sampling process. Within this scheme, the noisy latent $z_T$ can be transformed into a fully denoised latent $z_0$:

$$z_{t-1} = \sqrt{\frac{\alpha_{t-1}}{\alpha_t}} z_t + (\sqrt{\frac{1 - \alpha_{t-1}}{\alpha_{t-1}}} - \sqrt{\frac{1 - \alpha_t}{\alpha_t}})\epsilon_\theta, \quad t = T, ..., 1, \tag{14}$$

where $\alpha_t$ is a reparameterized noise scheduler.

## C    DETAILED DESCRIPTION OF ALGORITHM

**The Inflation of Attention Mechanism.**    Video generation models generally inflate text-to-image models. There exists an inflated 3D attention mechanism that enables the inter-frame interaction. Specifically, for the latent representation $z_t^i$ of source frame $f_t^i$, query features are derived from spatial features $z_t^i$, while key and value features are computed from spatial features of concatenated latent $[z_t^1, ..., z_t^N]$. The mathematical formulation can be as follows:

$$Q = W^Q \cdot z_t^i, K = W^K \cdot [z_t^1, ..., z_t^N], \quad V = W^V \cdot [z_t^1, ..., z_t^N].$$

**Particle Swarm Optimization.**    We use this algorithm in stage 2 to search for a video perturbation. PSO algorithm is inspired by the social behavior of birds flocking or fish schooling. There are some basic concepts in this algorithm. (1) Particles: these are the individual agents or solutions that explore the search space. Each particle represents the video perturbation vector in our experimental setting. (2) Swarm: The group of particles. It collectively explores the search space by sharing information. Typically, the size of the swarm should cover the search variable's dimension. (3) Position and Velocity: Each particle has a position that represents a potential solution and a velocity that dictates its movement through the search space. (4) Fitness Function: This function evaluates how good a particle's position is. The goal is to find the position that optimizes this function.

In the search process of the video perturbation vector, our search objective is the MSE loss between the current video's inversion latent and the anchor inverted latent optimized in stage 1. Video perturbation vectors represent particles. In our experimental setting, the video perturbation vector $V \in \mathbf{R}^{C \times F}$, which means we optimize a vector for every frame.

## D    HUMAN STUDY

This section clarifies how we do human evaluation. For frame consistency, we present normal edited videos and videos generated by our method and two baseline methods. We ask the raters "Please

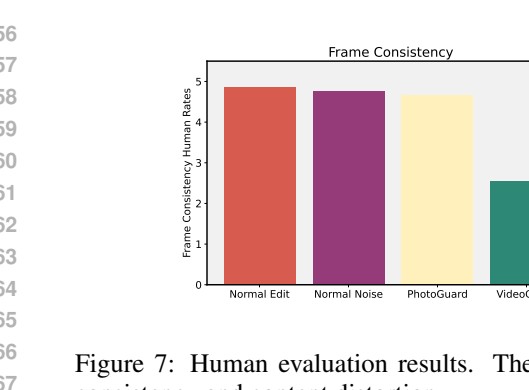

Figure 7: Human evaluation results. The figure reflects the human rates at the aspect of frame consistency and content distortion

rater the frame consistency from 1-5", where 5 represents the best frame consistency. For content quality, we ask the raters to rate the video content's naturalness from 1-5, where 5 represents the best. We choose 10 video text pairs to formulate a questionnaire and recruit 10 participants to annotate. We use the average score as the final result and it turns out that our method can effectively protect videos. We can obtain the aforementioned conclusion based on Frame Consistency dropping from 4.80 to 2.60, and Content Quality dropping from 3.50 to 1.60. Figure 7 visually displays the evaluation results.

# E  ABLATION STUDY

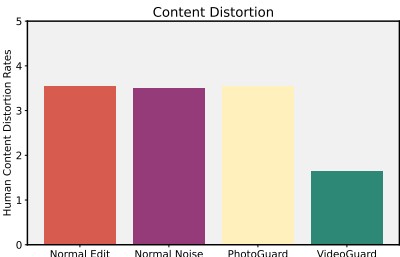

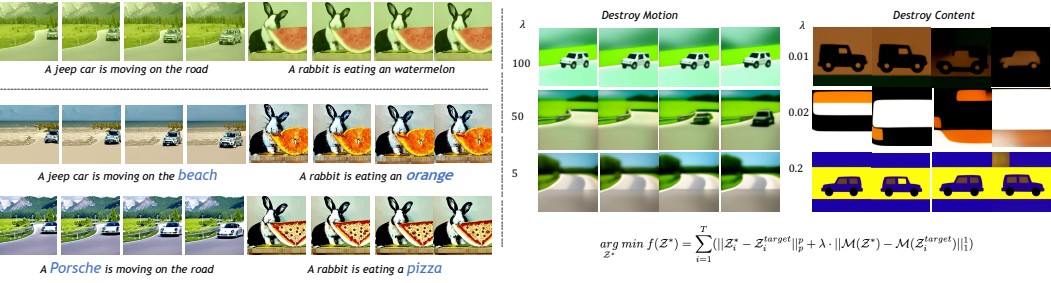

Figure 8: Ablation study. On the left, the first row represents the perturbed video frames without our method, and the following 2 rows are the edit results. The results show that random perturbed video can still be easily edited. On the right, different $\lambda$ represent different trade-offs between motion and content.

**Trade-off between content loss and motion loss.** In optimization stage 1, we optimize such an objective function $arg\min_{\mathcal{Z}^*} f(\mathcal{Z}^*) = \sum_{i=1}^{R}(||\mathcal{Z}_i^* - \mathcal{Z}_i^{target}||_p^p + \lambda \cdot ||\mathcal{M}(\mathcal{Z}^*) - \mathcal{M}(\mathcal{Z}_i^{target})||_1^1)$ to find a DDIM Inversion latent anchor. Actually, $\lambda$ is the trade-off between content destruction and motion destruction. Different $\lambda$ can result in different DDIM Inversion Latent anchors optimized in stage 1. To figure out the difference, we use the same prompt to guide the denoise process of different anchors. As shown in Figure 8 (right), when the motion loss takes the dominant proportion, the model will generate videos that lose motion dynamics while preserving the original video content. On the contrary, when the content loss takes the dominant proportion, the model will generate videos that lose content information while possessing motion dynamics.

**Perturbation vector optimization.** When doing optimization in video pixel space, we use the Particle Swarm Algorithm to search for a perturbation vector and fuse it into the original video. Figure 8 demonstrates the effectiveness of our method. When we fuse a random perturbation vector into the original video, namely a random color shift, the editing pipeline can still generate videos that are aligned to the target edit prompt and without any distortion.

**Advanced Algorithm.** To make protection more imperceptible and more natural, actually, we can add more naturalness constraints when doing perturbation optimization. Generally, we can use

PSNR, SSIM, and LIPIS to regularize, which can lead to a more natural immunized video. To be specific, let $V_{original}$ represent the original video, and $\delta_{video}$ represent the perturbation. We use $\mathcal{L}_{natural}$ to represent naturalness loss. For example, we can regard the common naturalness metrics like PSNR and SSIM as the naturalness loss:

$$o_1 = \text{PSNR}(V_{original}, V_{original} + \delta_{video}), \tag{15}$$

$$o_2 = \text{SSIM}(V_{original}, V_{original} + \delta_{video}), \tag{16}$$

and

$$L_{nat} = o_1 + o_2. \tag{17}$$

As such, the whole fitness function is

$$L_{total} = MSE(\text{DDIM Inversion}(V_{original} + \delta_{video}), z_{target}) + \mathcal{L}_{nat}. \tag{18}$$

where $z_{target}$ is the anchor latent optimized during stage 1.

## F  MORE RESULTS

To further demonstrate the efficacy of our method, we present more results.

### F.1  TRANSFERABILITY ACROSS DIFFERENT MODELS.

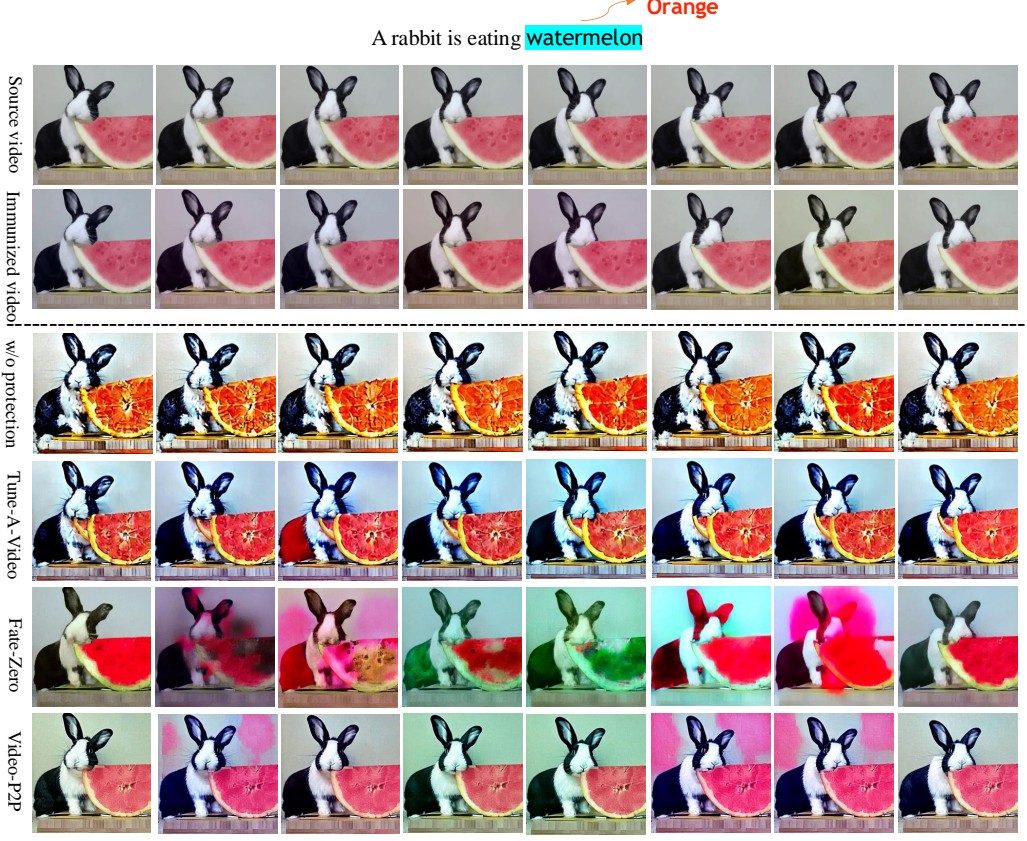

Figure 9: Transferability across different models. The first two rows are the source video and immunized video respectively. The third row represents the normal edit result without any protection. The following 3 rows show the protection effects of different models.

### F.2  PROTECTION RESULTS

Figure 10: Transferability across different models. The first two rows are the source video and immunized video respectively. The third row represents the normal edit result without any protection. The following 3 rows show the protection effects of different models.

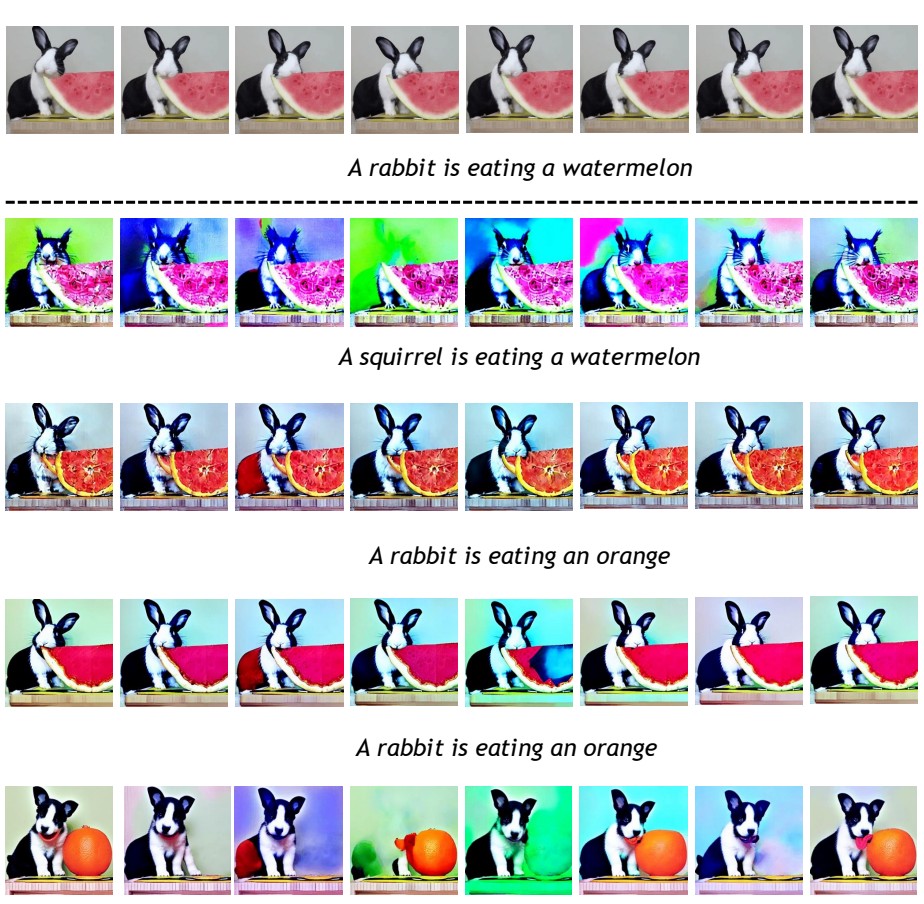

*A rabbit is eating a watermelon*

*A squirrel is eating a watermelon*

*A rabbit is eating an orange*

*A rabbit is eating an orange*

*A dog is eating an orange*

Figure 11: Different edit prompt results. The first row is the source video. The caption of the source video is "a rabbit is eating a watermelon", and the following lines represent different results of different instructions.

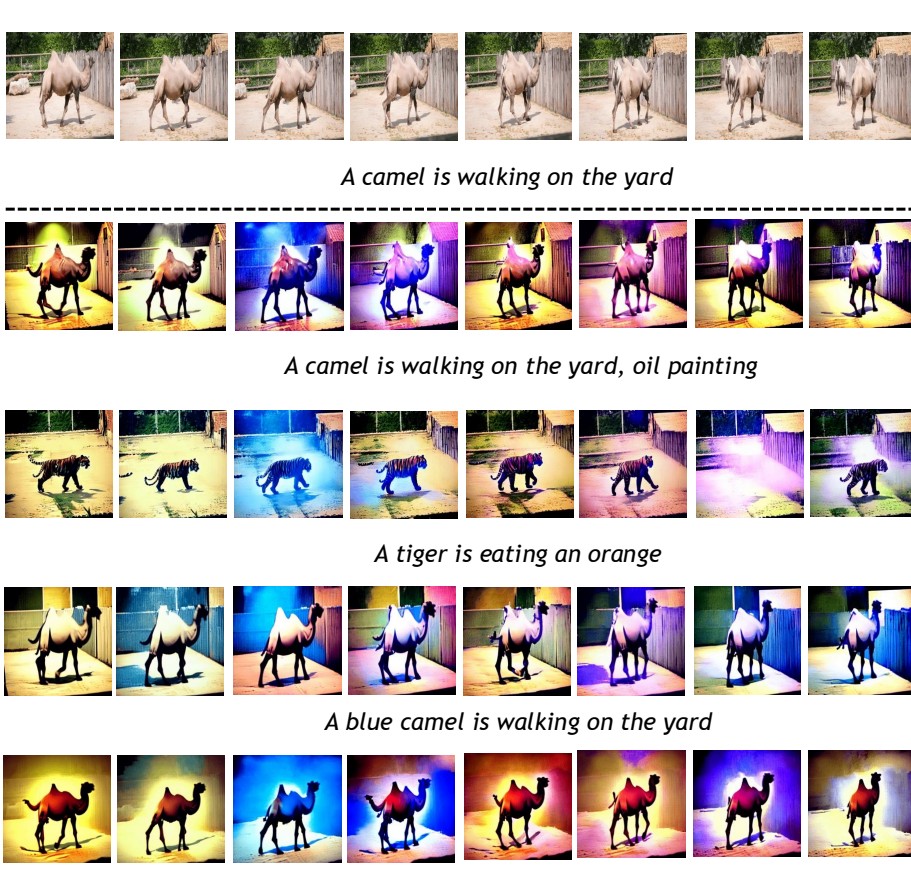

*A camel is walking on the yard*

*A camel is walking on the yard, oil painting*

*A tiger is eating an orange*

*A blue camel is walking on the yard*

*a camel is walking on the snow*

Figure 12: The first row is the source video. The caption of the source video is "a camel is walking on the yard", and the following lines represent different results of different instructions.

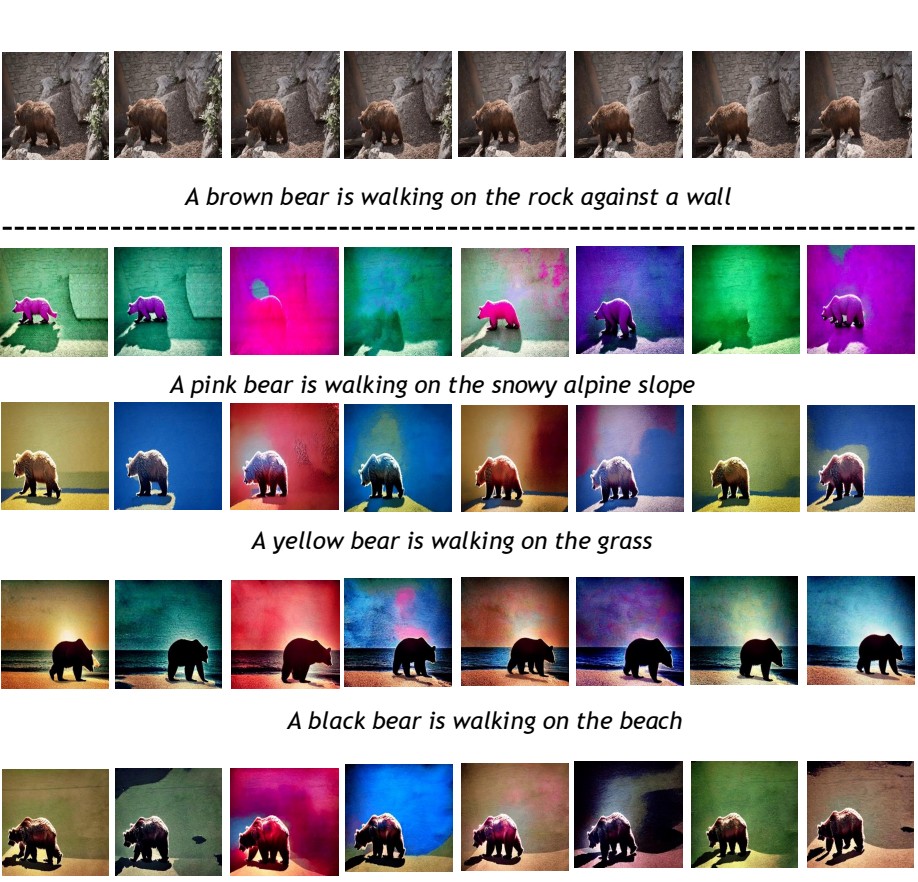

Figure 13: The first row is the source video. The caption of the source video is "a brown bear is walking on the rock against a wall", and the following lines represent different results of different instructions.

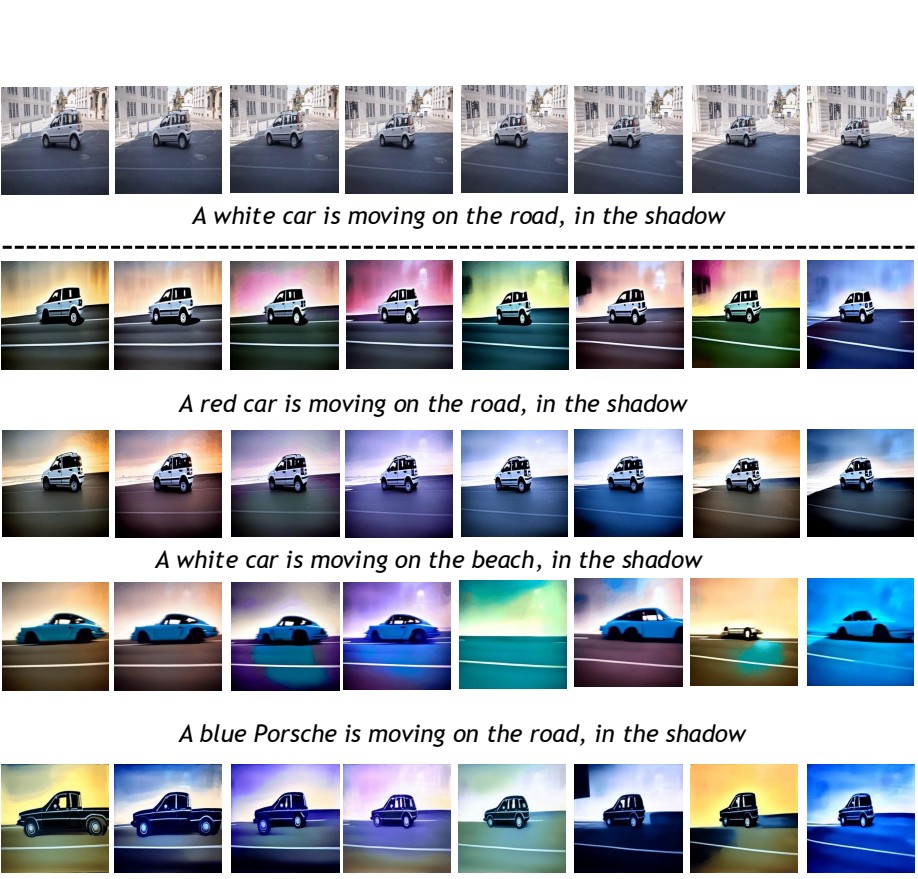

*A white car is moving on the road, in the shadow*

- - - - - - - - - - - - - - - - - - - - - - - - - - - - - - - - - - - - - - - - - -

*A red car is moving on the road, in the shadow*

*A white car is moving on the beach, in the shadow*

*A blue Porsche is moving on the road, in the shadow*

*A white car is moving on the road, in the shadow, Vangogh Style Painting*

Figure 14: The first row is the source video. The caption of the source video is "a white car is moving on the road, in the shadow", and the following lines represent different results of different instructions.

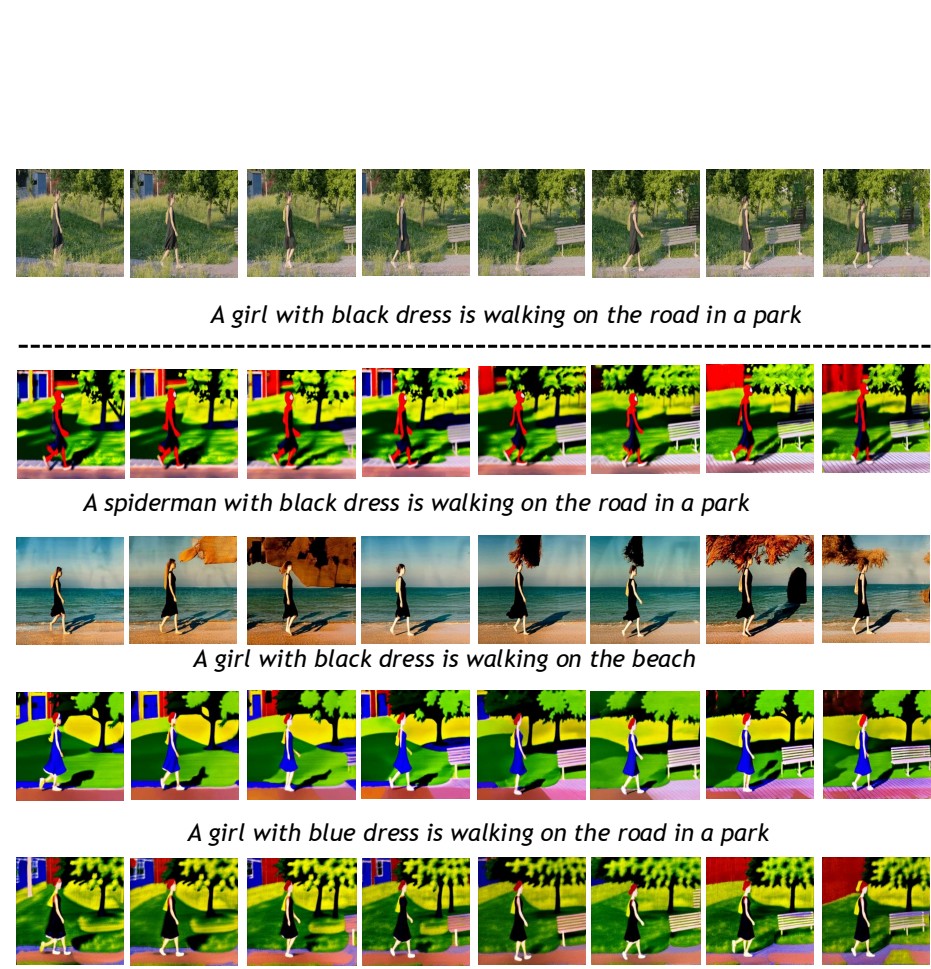

Figure 15: The first row is the source video. The caption of the source video is "A girl with a black dress is walking on the road in a park", and the following lines represent different results of different instructions.

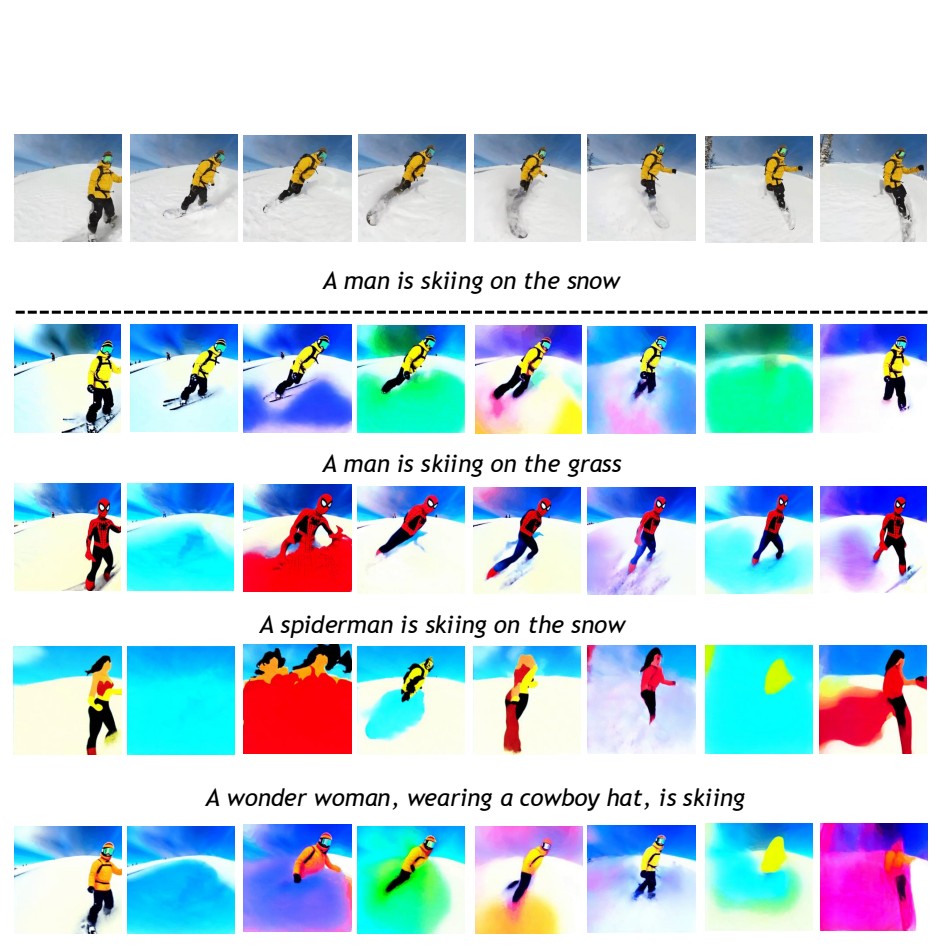

*A man is skiing on the snow*

*A man is skiing on the grass*

*A spiderman is skiing on the snow*

*A wonder woman, wearing a cowboy hat, is skiing*

*A man, wearing pink clothes, is skiing as sunset*

Figure 16: The first row is the source video. The caption of the source video is "a man is skiing on the snow", and the following lines represent different results of different instructions

