# OpenReview forum: "VIDEOGUARD: PROTECTING VIDEO CONTENT FROM UNAUTHORIZED EDITING"
_ICLR.cc/2025/Conference — Submitted to ICLR 2025_

### Official Review · Reviewer_Yvx1 · 2024-10-24

**Soundness:** 3
**Presentation:** 1
**Contribution:** 1
**Rating:** 5
**Confidence:** 4

**Summary:**

This work introduces a protection method called VideoGuard, designed to safeguard videos against unauthorized malicious editing. The optimization process involves analyzing all frames collectively, with the differences between frames extracted as motion information for optimization purposes.

**Strengths:**

1. This work adopts adversarial attack techniques from the field of image attacks into the video field to safeguard video editing methods that require the DDIM inversion step.

**Weaknesses:**

1. The video editing methods tested are quite outdated. Many contemporary video editing models, such as [1, 2, 3], do not require the DDIM inversion steps. In this case, does the method still remain effective?

2. The method appears to merely adopt techniques from the field of image attacks, but applied in a higher-dimensional space that includes the timestamp.

3. From Figure 3, it appears that the protection is ineffective, as the identities of Biden and Trump remain quite evident, with only minor color distortions present. To prevent the generation of these identities, methods for concept erasure, such as those described in [4, 5, 6, 7], could be considered. A brief discussion on concept removal in the related work section would be expected.

5. The layout of the main pipeline figure (Figure 2) is poorly arranged. The text and formulas should be enlarged for improved readability.

6. The citation format is incorrect, further diminishing readability. The authors have mistakenly used integral citations instead of non-integral citations. Please correct this.

---
[1] Consistent Video-to-Video Transfer Using Synthetic Dataset

[2] Video Editing via Factorized Diffusion Distillation

[3] Fairy: Fast parallelized instruction-guided video-to-video synthesis.

[4] Erasing Concepts from Diffusion Models

[5] Ablating Concepts in Text-to-Image Diffusion Models

[6] MACE: Mass Concept Erasure in Diffusion Models

[7] Separable Multi-Concept Erasure from Diffusion Models

**Questions:**

Please refer to the weaknesses.

If the authors adequately address my concerns during the rebuttal, I am open to adjusting my score.

---

### Official Review · Reviewer_LQZB · 2024-10-30

**Soundness:** 2
**Presentation:** 2
**Contribution:** 2
**Rating:** 3
**Confidence:** 4

**Summary:**

The paper introduces VideoGuard, a method designed to protect videos from unauthorized editing by generative diffusion models. Unlike prior efforts that primarily focus on images, VideoGuard specifically addresses the unique challenges posed by video content, including the sequential structure of frames and the presence of motion dynamics. The approach involves adding imperceptible perturbations across frames, which disrupt the generative model's operations without compromising the visual quality of the video. The paper presents a pipeline for implementing these perturbations and validates VideoGuard's effectiveness using various metrics.

**Strengths:**

This work represents the first attempt to prevent unauthorized manipulation of videos, paving the way for future research in this area.

**Weaknesses:**

**Novelty:** While this work is a pioneering attempt to counter unauthorized manipulation in video editing models, the proposed method demonstrates limited innovation. The two-stage learning approach lacks specific insights tailored to this task. Specifically, in both stage 1 and 2, the authors apply PGD to learn the perturbated latent code and protedcted videos. However, PGD is published in 2018, which is outdated. I suggest the authors to customize a new temporal attack method for the task.

**Motivation:** The authors propose that their method prevents unauthorized video editing, yet it essentially functions as a targeted attack, meaning the input video can still be manipulated. Namely, the goal of preventing unauthorized video editing is not fully achieved in this work. Additionally, the resulting videos align more closely with a target latent rather than the input prompt. Furthermore, the method requires full access to the model's architecture, presenting a significant limitation. Although cross-model results are reported in experiments, the method shows poor transferability to unseen models. In real-world scenarios, the transferability to unseen models and datasets is crucial. More transferability evaluations are expected to enhance the quality of this work.

**Methodology:** Stage 1 uses first-order difference to capture motion information, while Stage 2 aims to align the latent code of protected videos with an anchor code. However, these techniques are too simple and lack insights to the task of video protection. The authors do not fully use the temporal information in sequential signals.

**Experiments:** The method exhibits limited quantitative and qualitative improvements. In Table 1, its performance is only slightly better than the random noise baseline. This work fails to compare with SOTA attack methods, such as BSR [a] and ILPD [b]. The effectiveness of the qualitative results is not clearly demonstrated.

[a] K. Wang, X. He, W. Wang, and X. Wang. Boosting adversarial transferability by block shuffle and rotation. In Proc. IEEE Int’l Conf. Computer Vision and Pattern Recognition, 2024.
[b] Q. Li, Y. Guo, W. Zuo, and H. Chen. Improving adversarial transferability via intermediate-level perturbation decay. Proc. Annual Conf. Neural Information Processing Systems, 2023.

**Questions:**

What datasets are used in training and testing? It seems the testing data only comprises 80 text-video pairs. I concern that the data size is too limited to demonstrate the effectivenss of the proposed approach.

---

> ### Author Response · Authors · 2024-11-15
>
> Dear reviewer, we thank you for your detailed and constructive feedback on our paper. We have carefully considered all the comments, and we will try to address your confusion in the following parts.
>
> Firstly, we'd like to answer the question of dataset size. The dataset comprises DAVIS[1] videos, and some real-world videos downloaded from internet. We follow [2], [3], [4], [5], [6], their dataset ranges from 20 videos to 42 videos. We have tried to expand the experiments dataset, and finally set the size as 80. And then we will respond point by point.
> * Novelty and Methodology. We suppose there may exist some misunderstanding. The main contribution of our work is to formulate a perturbation optimization problem according to the characteristic of diffusion models and video signals. **Namely, the core of our contribution is how to formulate an optimization problem, not how to solve the problem.**  However, we indeed think you have pointed out a good future research direction, we can develop or utilize more powerful solvers and methods to solve the problem. Also, there might be other ways to build up optimization problems with other video information. **Again, our contribution lies in the process of proposing an optimization problem formulation tailed for video editing protection, and we demonstrated it can raises the cost of malicious editing.** PGD is proposed in 2018, PSO is proposed in 1957, but we are not focused on solvers. We sincerely hope our explanation may help you diminish your confusion, and you can read some **protection related papers** mentioned in our ***Related Work*** part in our paper.  [7] [8] [9]
> * Motivation. The motivation of our work, is to add a video-tailored perturbation into raw videos, so that the edit results of immunized videos contain distortion contents, which can be easily deemed unrealistic. We formulate a perturbation optimization problem, and use PGD to solve the problem in latent stage 1, and then use PSO to align the latent in video space. As you said, after analysis, we found we can transform it into a target attack problem. You also pointed out that there is a limitation that our work requires full access to the model's architecture. **Actually, we utilize a basic surrogate model based on SD1.5, utilize surrogate model for optimization, and we do not access to the target models in the experiment. The results of cross-model and cross-instruction show the transfer ability of our method.**  More and more powerful video generation techniques are emerging, especially merge-based editing methods and even recent flow-matching methods like SD3 and Flux 1.1, our protection method indeed can not cover all the unseen edit models just as mentioned in our **Limitation** part, but we indeed reveal that we can raise the cost of malicious editing, and we will dive deeper.
> * Experiments. As you can see in Fig. 3 and Fig. 5, with nearly imperceptible perturbation, the resulted videos will contain distortion content, especially color diffusion to the entire frame. As for the quantitative results, there are 5 viable unsupervised metrics in the video evaluation benchmark VBench, we calculate all the 5 metrics, and we can see that our method is obviously superior in the aspect of consistency, we'd like you refer to video generation papers like  [2] [3] [4] [5] [6] [7] [8] to see the value range. And the quality aspect is also lower than baseline methods which we think is an affiliate metric for generative task. Qualitative performance is more important in generative task, you can refer to the appendix, more results can demonstrate immunized videos will contain distortion content, or with subject diminishing after editing.  Suppose you look a video with distortion content and inconsistent subject, you can definitely regarded it as fake. Suppose you are a malicious editor, it will without doubt hinder you edit easily.
>
> We sincerely appreciate your effort of reviewing our work, and we sincerely hope the above explanation may help you have a deeper understanding of our work. Finally, if our discussion has changed your perspective on this work, we sincerely hope you would consider adjusting your score.
>
> ​																Best wishes,
>
> ​																2024. 11. 16
>
> [1] Learning unsupervised video object segmentation through visual attention.
>
> [2] FateZero: Fusing Attentions for Zero-shot Text-based Video Editing
>
> [3] Tune-A-Video: One-Shot Tuning of Image Diffusion Models for Text-to-Video Generation
>
> [4] Tokenflow: Consistent diffusion features for consistent video editing
>
> [5] VMC: Video Motion Customization using Temporal Attention Adaption for Text-to-Video Diffusion Models
>
> [6] GROUND-A-VIDEO: ZERO-SHOT GROUNDED VIDEO EDITING USING TEXT-TO-IMAGE DIFFUSION MODELS
>
> [7] Raising the cost of malicious ai-powered image editing
>
> [8] Prime: Protect your videos from malicious editing
>
> [9] Towards Reliable Verification of Unauthorized Data Usage in Personalized Text-to-Image Diffusion Models

---

> > ### Comment · Reviewer_LQZB · 2024-11-26
> >
> > Thank you to the authors for their detailed responses. The additional discussions and justifications have partially addressed my concerns. However, some of my concerns regarding novelty, methodology, and experiments remain insufficiently addressed. Therefore, I am maintaining my rating unchanged.

---

### Official Review · Reviewer_C7GU · 2024-11-06

**Soundness:** 2
**Presentation:** 2
**Contribution:** 2
**Rating:** 5
**Confidence:** 2

**Summary:**

This paper proposes VideoGuard to protect videos from unauthorized editing. VideoGuard consists of two-stage pipelines. In the first stage, the authors use DDIM inversion process to get an initial noise latent from the original video. Then, the method optimizes the latent so that the denoising from the latent will result in a disrupted video, and therefore the video editing will be unsuccessful.  In the second stage, the method tries to find a perturbation that after adding it on top of the source video it can make the inversion of the video close to the optimized latent from stage 1. In addition, the added perturbation is required to be imperceptible. Therefore, VideoGuard can produce a similar source video that is free of unauthorized editing.

**Strengths:**

1. The proposed approach is interesting and reasonable to achieve the goal of protecting videos from unauthorized editing.
2. VideoGuard is to protect the video in the video pixel space, and the enhanced video is similar to the source video with extra shield.
3. The method treats the video as a whole without needing to process each frame individually.

**Weaknesses:**

My major concern is the results of VideoGuard. Although VideoGuard is a reasonable approach, seems the performance is not very effective.

First of all, the video editing used seems not good. The edited videos are not very realistic from the examples shown in the paper. If the video editing method is not strong enough, it might be easy to protect the source video from effective editing.

Second, the protection result of VideoGuard is not good. As shown in the second row of Fig.3, VideoGuard fails to protect the source video. The immunized video is successfully changed by the prompt. Also, as shown in Fig. 5, the edited video protected by VideoGuard changed a lot, deviating a lot from the source video.

For the quantitative result, compared to the baseline without protection, the number did not make a large difference. I suppose the metric number will be very difference when comparing with and without protection.

**Questions:**

N/A

---

### Official Review · Reviewer_BCcu · 2024-11-08

**Soundness:** 3
**Presentation:** 2
**Contribution:** 2
**Rating:** 5
**Confidence:** 3

**Summary:**

The paper proposes VideoGuard to protect video content from unauthorized editing by diffusion-based video editing models.  VideoGuard introduces a two-stage optimization pipeline, including Optimizing DDIM Inversion Latent and Optimizing Video Perturbation.

**Strengths:**

1. This work regards the video and its inversion as a whole, instead of using an image-based approach and processing it frame by frame. It's reasonable.
2. Experimental results demonstrate the effectiveness of the two-stage approach.

**Weaknesses:**

1. The experiments are insufficient and do not introduce enough baseline methods to prove the superiority of the proposed method.
2. Lack of test results on more video editing methods.

**Questions:**

Please demonstrate the superiority of the proposed method through more sufficient experiments.

---

### Meta-Review · Area_Chair_FXy7 · 2024-12-20

**Metareview:**

The reviewers reached consensus on concerns regarding the effectiveness, novelty, and experimental validation of the proposed VideoGuard method. The paper's contributions are considered insufficient for acceptance on ICLR, and a rejection decision is recommended. The reviewers encouraged the authors to explore more innovative solution, strengthen their experimental design, and address the limitations related to generalizability and real-world applicability.

**Additional Comments On Reviewer Discussion:**

Authors only responded to ReviewerLQZB's comments. However, ReviewerLQZB still had concerns regarding novelty, methodology, and sufficiency of experiments.

---

### Decision · Program_Chairs · 2025-01-22

Reject